# A Modular Hepatitis E Virus Replicon System for Studies on the Role of ORF1-Encoded Polyprotein Domains

**DOI:** 10.3390/pathogens11030355

**Published:** 2022-03-15

**Authors:** Filip Cierniak, Rainer G. Ulrich, Martin H. Groschup, Martin Eiden

**Affiliations:** 1Institute of Novel and Emerging Infectious Diseases (INNT), Friedrich-Loeffler-Institut, 17493 Greifswald-Insel Riems, Germany; filip.cierniak@outlook.de (F.C.); rainer.ulrich@fli.de (R.G.U.); martin.groschup@fli.de (M.H.G.); 2Partner Site Hamburg-Lübeck-Borstel-Riems, German Centre for Infection Research (DZIF), 17493 Greifswald-Insel Riems, Germany

**Keywords:** hepatitis E virus, replicon, luciferase readout, ORF1, nonstructural polyprotein domains

## Abstract

Zoonotic hepatitis E virus (HEV) infection is an emerging cause of acute viral hepatitis in developed countries. Known reservoirs of zoonotic genotype 3 (HEV-3) are mainly pigs and wild boar, and to a lesser extent rabbits and deer. Rabbit hepatitis E virus (HEV-3ra) is prevalent in rabbits worldwide and represents a particular risk for zoonotic infection. Current understanding of the molecular mechanisms of HEV pathogenesis is incomplete, particularly due to the limited availability of efficient and reliable cell culture systems. In order to identify genomic regions responsible for HEV propagation in cell culture, we developed a modular chimeric reporter replicon system based on cell culture-adapted (Kernow-C1/p6 and 47832mc) and rabbit-derived HEV strains. Replication in HepG2 cells was monitored on the basis of a *Gaussia* luciferase reporter gene that was inserted in place of the open reading frame (ORF) 2 of the HEV genome. Luciferase activity of rabbit HEV-derived replicons was significantly lower than that of Kernow-C1/p6 and 47832mc replicons. Serial exchanges of defined ORF1 segments within the Kernow-C1/p6 replicon backbone indicated that HEV replication in HepG2 cells is not determined by a single domain but rather by an interplay of longer segments of the ORF1-derived nonstructural polyprotein. This implies that a specific combination of viral factors is required for efficient HEV propagation in cell culture.

## 1. Introduction

Hepatitis E virus (HEV) is a common cause of acute viral hepatitis worldwide [1]. The disease hepatitis E (HE) is usually subclinical and self-limiting; however, in some cases, fulminant hepatic failure is observed. Moreover, complications can arise in immunocompromised patients leading to chronic hepatitis mainly in solid-organ transplant recipients [2], or fetomaternal outcomes in pregnant women with mortality rates of up to 20% [3]. The virus contains a linear, single-stranded positive-sense RNA genome, which is capped at the 5′ end and polyadenylated at the 3′ end. The viral genome contains untranslated regions (UTR) at its 5′ and 3′ ends and three open reading frames (ORF). The ORF1 encodes a non-structural polyprotein that harbors the enzymatic activities of a methyltransferase, a protease, a macrodomain, a helicase, and an RNA-dependent RNA polymerase (RdRp), as well as a Y domain and a hypervariable region (HVR) of unknown function. ORF2 encodes the capsid protein and ORF3 encodes a small, multifunctional accessory protein [4].

Human infections were predominantly caused by strains of the *Orthohepevirus A* species of the *Hepeviridae* family: The genotypes HEV-1 and HEV-2 are transmitted solely between humans mainly by the fecal–oral route in endemic regions of Africa and Asia, where access to clean drinking water is often limited. In contrast, autochthonous HE cases in Europe and North America are mainly caused by zoonotic genotype HEV-3 with the main reservoir in pigs or wild boar and to lesser extent in rabbits and deer. Transmission is provoked by consumption of undercooked pork and meat products [5].

To monitor and elucidate transmission and clinical course of HEV infection, several animal models have been established. Although wild boar and pigs are highly susceptible to HEV [6] and can exhibit chronic infections [7], these animals do not exhibit any clinical symptoms beyond viremia and fecal virus shedding [8,9]. Similarly, chronic infections or fetomaternal complications could not be reproduced in HEV-infected non-human primates [10]. As alternative, European rabbit (*Oryctolagus cuniculus*) was developed as an alternative model for human HEV infection [11]. Rabbits can be experimentally infected with HEV-3 strains [12] and are associated with a specific subgenotype of HEV (HEV-3ra), which, in turn, can infect humans [13,14]. Furthermore, rabbits exhibit complications upon infection with HEV similar to humans, especially during pregnancy. Both chronic HEV infection [15] and poor fetomaternal outcomes [16] have been reported.

Many molecular aspects of HEV replication still remain unknown because the virus is challenging to efficiently propagate in cell culture. While some progress has been made in recent years, particularly with the development of cell culture models based on HEV-3 strains Kernow-C1 [17] and 47832c [18], the efficient cultivation of HEV in vitro is still limited to specific strains and cell lines. Both HEV strains were isolated from persistently infected patients and exhibit specific insertions within the HVR of the nonstructural protein encoding ORF1, which are critical for maintenance in cell culture [19,20]. In addition, optimized and well-adapted protocols allowed replication of HEV strains without particular insertions and production of high titers in cell culture [21]. However, the knowledge on conditions for HEV replication level in cell culture is still lacking.

Reverse genetics and subgenomic replicon systems in particular are powerful tools to elucidate the genetic characteristics responsible for efficient virus replication in vitro. In the context of HEV, notable examples include Nguyen et al. [22] and Cordoba et al. [23], demonstrating that HEV host specificity is governed not only by the ORF2-encoded capsid protein but also by the nonstructural protein of ORF1, or Tian and colleagues [24], who proved that intergenotypic recombination of fragments from the HVR or X region of ORF1 does not abolish HEV replication in vitro. However, few unique HEV strains have been used as an initial point for development of reverse genetics and replicon systems [25,26,27]. These strains are typically selected based on efficient growth in cell culture. In particular, the Kernow-C1/p6 system [19] has become a gold standard among HEV cell culture systems.

To evaluate the role of different segments of the ORF1-encoded polyprotein in a cell culture adapted replicon system, we built upon the p6/Luc replicon and assembled *Gaussia* luciferase expressing replicons based on HEV strains 47832mc [26], rabbit HEV strains rab52 [28], and rab81 [28,29]. As the reporter in these constructs substitutes 377 nucleotides of the ORF2/ORF3 overlap region, they cannot express ORF2- and ORF3-derived proteins. As a consequence, infectious particles are not formed [19]. This eliminates some of the complexity of the viral life cycle on one hand, but allows on the other for a more precise assessment of the replicase. Finally, we assembled chimeric replicons based on the p6 replicon backbone by partitioning ORF1 into three fragments and exchanging each of the fragments of p6 separately with the corresponding fragments from 47832mc, rab52, and rab81 strains.

## 2. Results

### 2.1. Construction of Luciferase Reporter Replicons Based on Different HEV-3 Strains

We constructed a set of novel luciferase reporter replicons based on HEV strains rab52, rab81, and 47832mc (Figure 1A). These strains represent different subclades of HEV-3, with pairwise ORF1 amino acid sequence identities between 84.9% and 92.2% (Table 1). The overall replicon architecture is based on the p6Luc replicon [19], which contains an ORF-encoding *Gaussia* luciferase in place of the first 377 nucleotides of ORF2 (positions 5359 to 5735). This deletion disables expression of functional capsid protein and removes all but the first eleven nucleotides of ORF3. The luciferase reporter gene was inserted in equivalent positions of rab52-, rab81-, and 47832mc-derived replicons.

All constructs were generated with the same plasmid backbone (pMK2) and contain a T7 promoter upstream for in vitro transcription, a 26 nucleotide polyA tail followed by a SwaI recognition site for DNA linearization prior to transcription (Figure 1B). For consistency, the p6/Luc replicon was reassembled in the same vector. Additionally, a negative control replicon was constructed by introducing three point mutations into the sequence encoding the conserved GDD motif of the RdRp, resulting in an inactivated GAA mutant. Finally, the 5′ ends of rab81 and rab52 were adapted to enable in vitro transcription with T7 RNA polymerase. The resulting replicons are referred to as p6LucA26, p6GAALucA26, 47832mcLucA26, rab52LucA26, and rab81LucA26 (Appendix A).

### 2.2. Rabbit HEV-Based Replicons Generate Low Luciferase Activity

After electroporation of HepG2 cells with in vitro transcribed (ivt) RNAs, reporter gene expression was detectable for all four constructs (Figure 2). The luciferase activity for the RdRp GAA mutant replicon remained at mock control background level. Luciferase activity was detected for ivt RNAs of each replicon construct from the first day after transfection and typically reached peak expression on the second or third day. Hereafter, the signal decreased but remained detectable for approximately five days for the rabbit HEV replicons and at least a week for p6 and 47832mc replicons (Figure 2). For parental p6LucA26 in particular, approximately 3 × 10^4^ relative light units (RLU) were measured on the first day. On day three after transfection, peak luciferase activity reached 1.5 × 10^6^ RLU and decreased to approximately 10^4^ RLU one week after transfection. In comparison to p6, 47832mc yielded higher luciferase activity (maximum: 3.6 × 10^6^ RLU at day three) and retained slightly higher expression levels in the long term (5.9 × 10^4^ RLU vs. 2.2 × 10^4^ RLU at day seven). In contrast, both the rab81- and rab52-based replicons displayed only low luciferase levels, with peak activities of 227 RLU at day three and 203 RLU at day two, respectively. Both replicon activities dropped to base line over seven days and were no longer distinguishable from the mock background. It is notable that the peak maxima were all reached by day three regardless of whether the respective replicon was based on cell culture adapted or non-adapted strains.

### 2.3. Construction and Evaluation of Chimeric ORF1 HEV Replicons

In order to investigate the different activities of the HEV replicons, the influence of corresponding ORF1-derived nonstructural polyproteins was analyzed by generation of chimeric ORF1 replicons based on the p6-Luc backbone. For this purpose, the ORF1 was divided into three fragments spanning the 5′-UTR, and coding region for methyltransferase, Y-domain, putative protease (MYP, corresponds to nucleotide positions 1–2139 in p6), coding region for hypervariable region, X-domain, helicase (VXH, corresponds to nucleotide positions 2144–4059 in p6) as well as RdRp encoding region and junction region (RJ, corresponds to nucleotide positions 4064–5347 in p6) (Figure 3A). The fragments are connected to each other by tetranucleotides, which are highly conserved across HEV-3 genomes (positions in p6: 2140–2143 “GGUC”; 4060–4063 “UGCC”; 5348–5351 “AUGG”) and consequently serve as convenient overhangs for a type IIS restriction-ligation-based recombination strategy. The fragments were inserted into the respective p6LucA26 constructs, resulting in nine chimeric constructs (Figure 3A).

After transfection of HepG2 cells, luciferase signals were monitored over a period of seven days (Figure 3B). In general, all tested replicons with exchanged subgenomic fragments produced measurable luciferase signals, higher than those observed for mock transfected and RdRp knockout (p6GAALucA26) controls. First, the positive control p6LucA26 showed a luciferase readout of 10^4^ RLU already on day one, and reached the maximum value of 1.5 × 10^6^ at day three, followed by a decrease to just below the initial value (9.8 × 10^3^ RLU) over the course of the observation period. Substitution of the p6 parental domains by those of 47832mc resulted in similar kinetics of luciferase activity compared to the p6 control but with slightly higher values: Exchange of MYP domain led to increased activity with a value of 2.1 × 10^6^ RLU already on day two followed by maintenance of increased values from day five until the end of the observation period. Insertion of VXH.47832mc domain led to higher values at day four (1.1 × 10^6^ RLU vs. 8.1 × 10^5^ RLU) and chimeric RJ.47832mc replicon peaked at day three (1.8 × 10^6^ RLU vs. 1.5 × 10^6^ RLU). In contrast, while the RJ fragments from rab52 or rab81 demonstrated high activity within the respective p6 chimera, chimeric replicons with subgenomic fragments MYP and VXH from both strains induced a dramatic reduction on the replicon activity at about two orders of magnitude compared to parental p6. Insertion of MYP.rab52 or MYP.rab81 showed peak values at day four of about 7.5 × 10^3^ and 1.3 × 10^4^ RLU compared to 8 × 10^5^ RLU for the p6 replicon. Similarly, VXH.rab52 and VXH.rab81 domains caused a reduction to about 7 × 10^3^ RLU compared to 8 × 10^5^ RLU for the p6 replicon. Furthermore, apart from a delayed increase, the local maximum of the luciferase expression curve is not as pronounced with the chimeric replicons. Rather than the distinct expression peak exhibited by p6 on day three post transfection, the curve plateaus from day three until day five, followed by a slight decrease between day five and day seven of the experiment. Interestingly, although VXH domains of rab52 and rab81 have low sequence identity (Table 2), the luciferase activity curves of both chimeric replicons are almost identical.

## 3. Discussion

Here, we describe the establishment of a modular luciferase replicon system based on subgenomic fragments of different HEV strains. Starting with the re-assembly of the parental p6 reporter replicon and the generation of the p6 GAA RdRp knockout mutant, we further assembled three novel replicons based on HEV-3c strain 47832mc, HEV-3ra strain rab52, and HEV-3 (putative novel subgenotype) strain rab81. Luciferase expression curves of all constructs are characterized by a distinct peak, followed by a significant decrease of luciferase activity. This behavior is consistent with the original description of the p6/Luc replicon [19] and is attributed to the absence of capsid protein synthesis due to insertion of the *Gaussia* luciferase coding sequence. As previously observed by Nguyen et al. [22], the peak of luciferase activity occurred earlier in our experiments, likely due to the difference in transfection protocols or cell lines. The cell culture isolate 47832mc-derived replicon demonstrated similar and even higher luciferase activity compared to the positive control p6LucA26. In contrast, the replicons based on rab52 and rab81 demonstrated markedly lower activity. The similarity of the luciferase kinetics for these two replicons is surprising, as the sequence identity between both replicons is not particularly high (77.8% RNA sequence identity for the complete replicon sequences). Therefore, this similar activity cannot be explained by the close similarity of the two strains and needs further investigation. Although the luciferase expression kinetics of each full-length replicon may show the level of adaptation to growth in cell culture, this does not explain the replication efficiency of a given strain. In order to elucidate the influence of different ORF1 related subgenomic fragments on total replicon activity, we next decided to generate chimeric replicons of the different strains.

Substitution of N-terminal region, termed ‘MYP’ for the domains it contains, methyltransferase, Y-domain, and protease, resulted in a decrease of the luciferase expression peak by two orders of magnitude when MYP fragments of rab52 or rab81 were used. On the other hand, the activity of the construct containing the MYP fragment of 47832mc substantially exceeded the unmodified replicon. While the expression peak was only slightly higher, it was reached a day earlier. Furthermore, on the first day of the observation period, the MYP.47832mc chimeric replicon had already reached a luciferase output 36-fold higher than the p6LucA26 replicon. This property is not fully reflected by the 47832mcLucA26 replicon, which exceeds p6LucA26 but not to this extent. The difference of almost three orders of magnitude between the “best” and the “worst” MYP-chimeric construct (MYP.47832mc and MYP.rab52, respectively) represents the largest single difference between any two chimeric replicons and stresses the importance of this fragment for HEV replication in vitro. This is interesting, especially in light of recent reports of otherwise unremarkable HEV strains, which still grow efficiently in cell culture [21]. It poses the question which functions of this region cause this impressive difference in luciferase expression between the different MYP fragments. Firstly, the methyltransferase is involved in RNA capping [30], which is directly related to and crucial for translation of newly synthesized HEV RNA [31] in the host cell. The methyltransferase and protease domains are also relevant for suppression of retinoic acid inducible gene I (RIG-I) -mediated detection of the viral RNA and subsequent activation of the interferon pathway [32]. The protease domain has chymotrypsin-like cleavage and inhibition patterns, has been implicated in processing of the viral proteins [33], and inhibits the host interferon pathway [32,34]. The function of the Y-domain remains unknown, although a mutagenesis screen has revealed critical amino acid residues and secondary structure motifs within the sequence [35]. The 5′-terminal end of the HEV genome interacts with the viral RNA polymerase to facilitate genome replication [36]. In-depth investigation of this sequence and detailed examination of each subdomain could prove valuable.

The middle region, termed VXH for its HVR, X-domain, and helicase part, was of particular interest. The p6 replicon contains an insertion within the HVR and three amino acid exchanges within the X-domain, which are required for efficient growth in cell culture. Shukla et al. described modified constructs without insertion, which caused up to 50-fold decreased luciferase expression, and without the three X-domain mutations, which reduced the output by a factor of 2.3 to 5.1 [19]. This is in accordance with the decrease in the observed luciferase signal when the VXH fragments of rab52 or rab81 were used here. Interestingly, the expression kinetics of the VXH.rab52 and VXH.rab81 chimeric replicons were almost identical. This is surprising because the overall sequence identity between the two VXH fragments is rather low. In fact, VXH.rab52 contains the characteristic HEV-3ra insertion, while VXH.rab81 does not. Nevertheless, both fragments appear to have an almost identical effect on the activity of the replicon. On the other hand, VXH.47832mc only slightly affects the luciferase expression curve. This is consistent with the similarity of the expression kinetics of p6LucA26 and 47832mcLucA26. The VXH.47832mc fragment is derived from a cell culture adapted strain and also contains an insertion in the HVR, which is critical for growth in cell culture [20]. The influence of these insertions on virus replication is still not fully understood, though some evidence exists that they contain nuclear localization signals [20,37]. However, nuclear localization of the polyprotein by itself does not appear to improve replication in cell culture [37]. Notably, Scholz et al. demonstrated that exchanging the insertion in 47832mc for the insertion of p6 attenuates the virus beyond recovery [20]. The sequence length of VXH fragment exchanged in this work is larger, which may account for the comparably lower difference between the replicons. On the other hand, it is possible that the effect of the insertion is dependent on the sequence context of the HVR it is embedded in. When comparing the two strains, it is evident that the insertion of p6 is shorter by 15 nucleotides (171 nucleotides vs. 186 nucleotides), but the overall length of the HVR + insertion is identical between both strains (Appendix A).

The C-terminal region, RJ, contains the RdRp and the junction region. The subgenomic promoter, which controls the transcription of the subgenomic RNA with the reporter gene, is located at the 3′-end of the RdRp coding region. Of all fragments examined, RJ had the weakest effect on replication level, measured by luciferase signal release. RJ.47832mc increased the expression slightly, which is intriguing, given the 99% sequence identity on the amino acid level to RJ.p6 (Table 2). In contrast, RJ.rab81 decreased it by a small amount. Interestingly, RJ.rab52 yielded the highest values on day two, before dropping back to the level observed with the unmodified replicon, and subsequently reach the luciferase expression levels similar to the RJ.rab81 chimera. Both RJ.rab52 and RJ.rab81 contain the K1634 variant of RdRp (ORF1-polyprotein, nomenclature based on HEV-1; the actual amino acid positions are 1706 in Kernow-C1/p6, 1663 in rab52, and 1641 in rab81), an amino acid exchange that is known to increase replication compared to the G1634 residue originally found in p6 [38]. Despite this, the RJ regions of rab52 and rab81 as a whole do not increase luciferase signal compared to the RJ.p6 fragment. The subgenomic promoter within the fragment may further influence the luciferase expression curve [36,39]. For detailed work on the function of the RdRp, it is advisable to keep the subgenomic promoter sequence unchanged.

Future work will expand the scope of both host cell lines and model replicons. Additional replicons based on different HEV strains, either cell culture adapted or non-adapted wildlife- or human-derived, as well as the corresponding chimeras, should reveal in more detail viral factors that are required for growth in vitro. On the other hand, different host cell lines will show how host factors interact with different HEV genotypes. Beyond HepG2, other cell lines, such as HuH-7, PLC/PRF/5, and A549, have shown promise within some cell culture protocols. Indeed, in light of our results with the 47832mc replicon, the accompanying A549 subclone D3 [40] appears to be a promising system for future experiments.

## 4. Materials and Methods

### 4.1. HEV Strains, Plasmids, and Cell Culture

Plasmids containing the full-length sequences of Kernow-C1/p6 (GenBank acc. JQ679013) and p6-Luc were generously provided by Patricia Farci (National Institutes of Health, Bethesda, MD, USA). Clone 47832mc (GenBank acc. MN756606) was a kind gift from Reimar Johne and Johannes Scholz (Federal Institute for Risk Assessment, Berlin, Germany). Synthetic DNA fragments encompassing the rab52 (GenBank acc. KY436898) genome were produced by Eurofins Genomics (Ebersberg, Germany). Synthetic DNA fragments encompassing the rab81 (GenBank acc. MT920909) genome were produced by Twist Bioscience (San Francisco, CA, USA). The 5′ ends of rab81 and rab52 were adapted as follows: The first nucleotide of rab81 was mutated to G to enable in vitro transcription with T7 RNA polymerase. The 5′ end of rab52 (KY436898) is truncated and was therefore extended and adapted to the consensus sequence of HEV-3ra 5′ UTRs. Plasmids containing the *Gaussia* luciferase reporter gene for insertion in rab52, rab81, and 47832mc replicons were ordered from Twist Bioscience (San Francisco, CA, USA).

HepG2 cells were purchased from CLS (Eppelheim, Germany) and grown in DMEM (Gibco 52100, ThermoFisher, Waltham, MA, USA), supplemented with NaHCO_3_ and Na-pyruvate, and 10% fetal bovine serum, including 10 mg/L gentamicin (PAN-Biotech, Aidenbach, Germany), and 250 mg/L amphotericin B (PAN-Biotech, Aidenbach, Germany) in cell culture flasks with vented caps (Corning, Corning, NY, USA) in a humidified incubator with 5% CO_2_.

### 4.2. PCR and Cloning

Subgenomic fragments were selected as follows: First, the genomes were scanned for conserved tetranucleotides to serve as recombination sites. Subsequently, the candidate sites were narrowed down to represent the functional domains [41] of the HEV-encoded proteins. Finally, optimal overhangs were selected using the data on overhang ligation fidelity by Potapov et al. [42] in order to maximize assembly efficiency.

The fragments were amplified from plasmid templates using Q5 High Fidelity DNA Polymerase (NEB, Ipswich, MA, USA) or Phusion Hot Start Flex DNA Polymerase (NEB, Ipswich, MA, USA). PCR products were gel purified using the Gel extraction Kit (Qiagen, Hilden, Germany) or the Wizard SV Gel and PCR purification Kit (Promega, Madison, WI, USA). All gel extractions were done by cutting out bands from a Sybr Safe (ThermoFisher, Waltham, MA, USA) stained agarose gel using blue light transillumination. Purified fragments were either used directly for assembly or first cloned with PCR cloning plasmid pMiniT 2.0 using the PCR cloning Kit (NEB, Ipswich, MA, USA). The resulting plasmids were amplified in *Escherichia coli* [43] and verified by restriction digest and sequencing analysis. Plasmid DNA was purified using the QIAprep Spin Miniprep Kit (Qiagen, Hilden, Germany) or the PureYield Plasmid Miniprep System (Promega, Madison, WI, USA) and digested using Esp3I (NEB, Ipswich, MA, USA) or Anza BpiI (ThermoFisher, Waltham, MA, USA), according to the manufacturer’s instructions and column-purified or gel extracted as described above. Fragments were ligated using T4 DNA ligase (NEB, Ipswich, MA, USA) and used to transform *E. coli* DH5α cells. All inserts were sequenced using the dideoxy chain termination method (Sanger sequencing) at Eurofins (Ebersberg, Germany). Backbone plasmid pMK2 was constructed by PCR amplification of the backbone of pET-19b (Novagen) and inserting the kanamycin resistance gene from pcDNA3-EGFP (a gift from Doug Golenbock; Addgene Plasmid #13031). Cloning and sequencing primer sequences are provided in the Appendix A. A listing with descriptions of all replicon plasmids (Appendix A) and cloning intermediates (Appendix A) is also provided in the Appendix A.

### 4.3. In Vitro Transcription and Transfection

Template DNA was generated by transforming *E. coli* with the appropriate, replicon-containing plasmids and growing an overnight culture in LB medium with 50 µg/mL kanamycin. Plasmids were purified using the QIAprep Spin Miniprep Kit (Qiagen, Hilden, Germany). The plasmids were linearized using SwaI (NEB, Ipswich, MA, USA). Linearized plasmids were purified using the Wizard SV Gel and PCR purification Kit (Promega, Madison, WI, USA) and quantified using a Nanodrop 2000c spectrophotometer (ThermoFisher, Waltham, MA, USA). Replicons were transcribed from 1 µg linearized DNA template using the HiScribe T7 ARCA mRNA Kit (NEB, Ipswich, MA, USA) according to the manufacturer’s instructions. If needed, the reactions were scaled as appropriate. ivtRNA was subsequently purified using the provided LiCl solution, and resuspended in nuclease-free water. Success of transcription reactions was confirmed by Nanodrop measurement and by running the RNA preparations on a 1% agarose gel. For transfection, the purified RNA was diluted to 125 ng/µL in a volume of 40 µL in nuclease-free water. A total of 1 µL was used for agarose gel analysis of the transfection mix. HepG2 cells were trypsinized and washed with and resuspended in OptiMEM (ThermoFisher, Waltham, MA, USA). After counting using a Neubauer chamber without trypan blue exclusion, the suspension was adjusted to 3.3 × 10^6^–3.5 × 10^6^ cells/mL. A total of 360 µL of this suspension was mixed with the diluted ivtRNA and transferred into a 4 mm electroporation cuvette (VWR, Radnor, PA, USA). The cells were then immediately electroporated using a Square Wave pulse for 20 ms at 300 V in a GenePulser XCell electroporation device (Bio-Rad, Hercules, CA, USA). After at least ten minutes of regeneration inside the cuvette, the cells were transferred into 1.2 mL of DMEM. This suspension was then seeded in four wells of a 96-well plate at 100 µL per well and incubated in a humidified incubator at 37 °C with 5% CO_2_. Medium was exchanged daily and the supernatants were collected and stored frozen at −80 °C. Each transfection was done at least three times.

### 4.4. Luciferase Reporter Assay

Initially, luciferase activity was measured using the Pierce Gaussia Flash Assay Kit (ThermoFisher, Waltham, MA, USA) and an infinite M200 Pro plate reader (Tecan, Männedorf, Switzerland). A total of 20 µL supernatant of cultures transfected with *Gaussia*-luciferase expressing HEV replicons were transferred to black 96-well plates (ThermoFisher, Waltham, MA, USA). Substrate working solution was prepared by diluting coelenterazine according to the instructions of the kit. A total of 50 µL working solution was injected per well, followed by 1s of shaking and light measurement with 0.5 s integration time. Alternatively, we diluted coelenterazine (Carl Roth, Karlsruhe, Germany) in PBS with 5 mM NaCl according to [44]. For statistical analysis, each group was compared to the positive control, p6LucA26, in an unpaired, two-tailed *t*-test. *p* values were adjusted for multiple testing using the FDR method [45]. Data analysis and visualization was done using R [46] with packages readxl [47], tidyr [48], tibble [49], rstatix [50], dplyr [51], ggplot2 [52], ggpubr [53], and ggh4x [54] in RStudio [55].

## 5. Conclusions

We established a modular HEV replicon system and demonstrated that the different luciferase outputs of chimeric replicons reflect the activity of their donor replicons. Our work provides an easy and efficient procedure to identify viral factors required for replication in vitro as well as potential bottlenecks. Beyond replicons, this basic approach is transferable to infectious cDNA clones and to study and compare the effect of different viral sequences in vivo.

## Figures and Tables

**Figure 1 pathogens-11-00355-f001:**
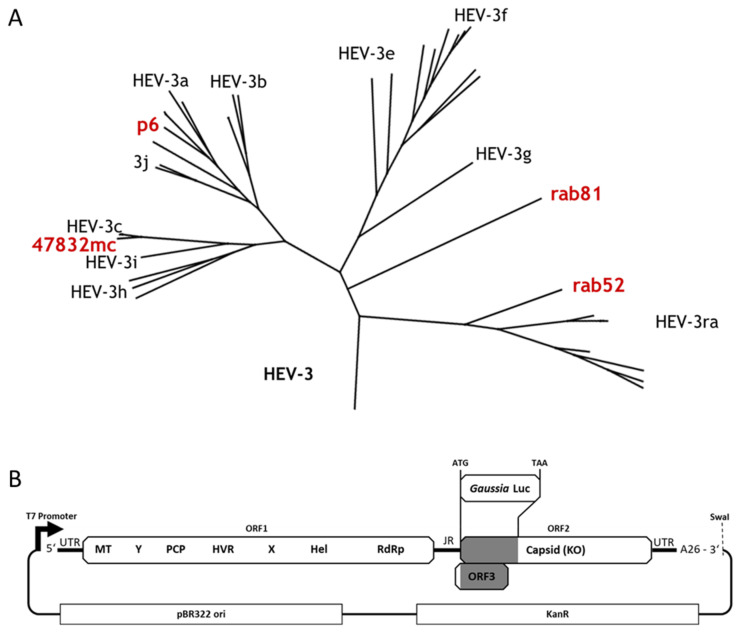
(**A**) Phylogenetic overview of the model strains used in this work. Depicted in red are HEV-3 strains p6 [17], 47832mc [26], rab52 [28], and rab81 [28,29]. The tree was constructed in Geneious using the Maximum Likelihood method. Subgenotype and strain labels were then added manually. (**B**) Schematic overview of parental replicon plasmid constructs. The sequence of each replicon was inserted downstream of a T7 promoter for in vitro transcription. A *Gaussia* luciferase (Luc) reporter gene was inserted at the start codon of ORF2, deleting 377 nucleotides of ORF2, or 380 nucleotides in the case of the rab81-based replicon, and almost the entirety of ORF3. Each construct contains a polyA tail of exactly 26 nucleotides and a SwaI restriction site directly downstream of the polyA tail for linearization prior to transcription. All constructs were cloned with the same minimal pMK2 plasmid backbone, which contains a pBR322-derived origin (ori) of replication, and a kanamycin resistence gene (KanR). MT = methyltransferase; Y = Y-domain; PCP = papain-like cysteine protease; HVR = hypervariable region; X = X-domain (macrodomain); Hel = helicase; RdRp = RNA-dependent RNA polymerase.

**Figure 2 pathogens-11-00355-f002:**
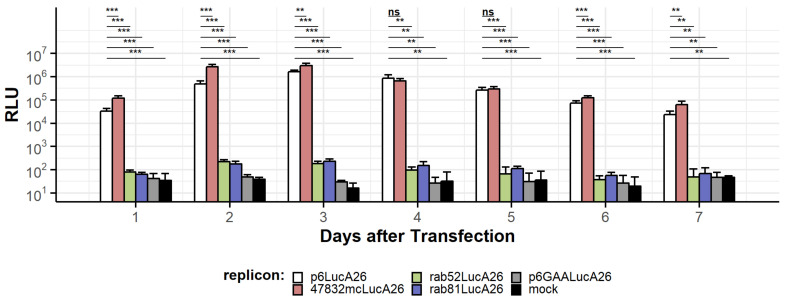
Luciferase activity of four parental HEV replicons in comparison to mock-transfected control (H_2_O) and inactivated RdRp GAA mutant replicon based on p6LucA26. HepG2 cells were transfected with replicon ivtRNA by electroporation and seeded across four wells of a 96-well plate as technical quadruplicate. Replication was estimated by measuring total luciferase activity (relative light units; RLU) of each replicon. Error bars indicate standard deviation and asterisks indicate significance level (*** *p* ≤ 0.001; ** 0.001 < *p* ≤ 0.01; ns 0.05 < *p* ≤ 1). The data depict a representative experiment of three separate transfections.

**Figure 3 pathogens-11-00355-f003:**
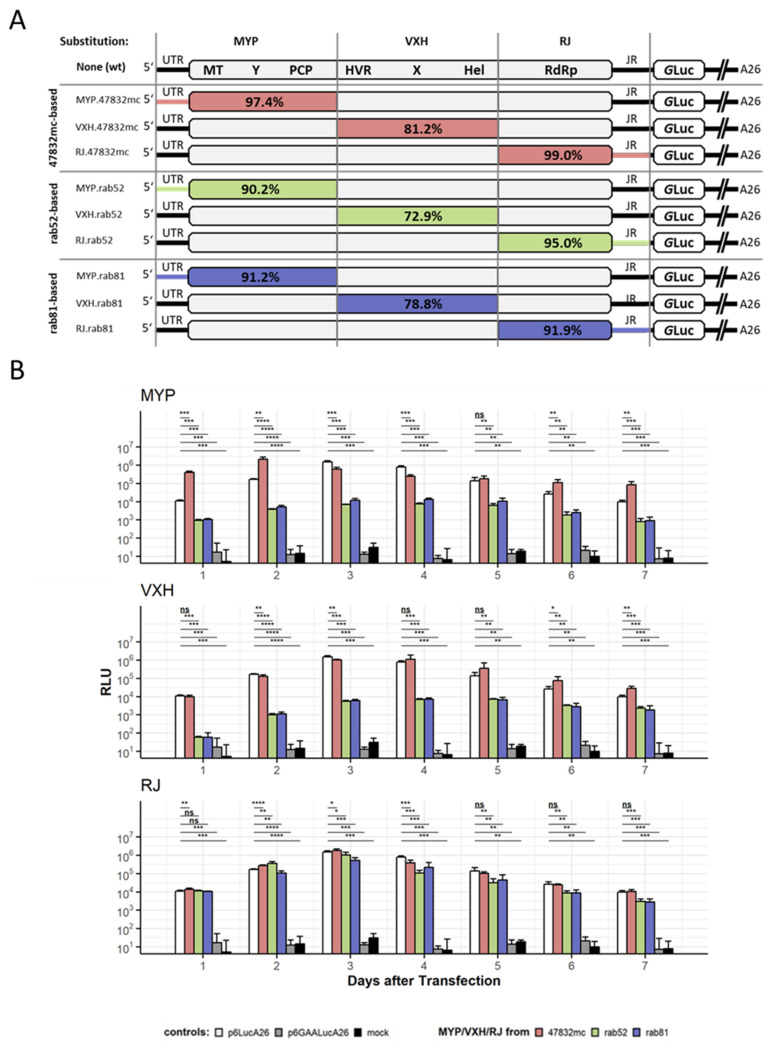
(**A**) Structure of nine chimeric replicons that were constructed using p6LucA26 as backbone. Subgenomic fragments containing partial sequences of ORF1 (MYP, nucleotide positions 1–2139 in p6; VXH, nucleotide positions 2144–4059 in p6; RJ, nucleotide positions 4064–5347 in p6) were replaced with the corresponding sequences from HEV strains 47832mc, rab52, and rab81. Percentages annotated in the highlighted fragments represent pairwise amino acid sequence identity of the encoded proteins with the corresponding ones encoded by parental p6 replicon. Multiple amino acid sequence alignments of each ORF1-encoded protein segment are included in the Appendix A. (**B**) HepG2 cells were transfected with replicon ivtRNA by electroporation and seeded across four wells of a 96-well plate as technical quadruplicates. Replication was estimated by measuring total luciferase activity (RLU) of each replicon. Error bars indicate standard deviation; asterisks indicate significance level in reference to p6LucA26 (**** *p* ≤ 0.0001; *** 0.0001 < *p* ≤ 0.001; ** 0.001 < *p* ≤ 0.01; * 0.01 < *p* ≤ 0.05; ns 0.05 < *p* ≤ 1). The data depict a representative experiment of three separate transfections. Note that the controls (p6LucA26, p6GAALucA26, and mock) are shown in each plot for illustrative purposes. The data depicted are derived from a single experiment with all chimeric replicons and one group of controls.

**Table 1 pathogens-11-00355-t001:** Pairwise nucleotide and amino acid sequence identities of the four reporter replicon sequences and the corresponding ORF1-encoded proteins.

Replicon Nucleic Acid Sequence Identity				ORF1 Nucleic Acid Sequence Identity				ORF1 Amino Acid Sequence Identity			
	rab81	rab52	47832mc		rab81	rab52	47832mc		rab81	rab52	47832mc
rab52	77.7%			rab52	74.4%			rab52	86.7%		
47832mc	78.4%	77.5%		47832mc	75.1%	73.5%		47832mc	86.5%	85.3%	
p6	78.5%	77.1%	83.5%	p6	75.5%	73.4%	81.3%	p6	87%	84.9%	92.2%

**Table 2 pathogens-11-00355-t002:** Pairwise nucleotide/amino acid sequence identities of the subgenomic fragments/protein domains in the chimeric constructs/proteins.

**Nucleotide Sequence Identity**											
**MYP**				**VXH**				**RJ**			
	rab81	rab52	47832mc		rab81	rab52	47832mc		rab81	rab52	47832mc
rab52	76.9%			rab52	68.6%			rab52	79%		
47832mc	79.1%	78.4%		47832mc	67.7%	63.7%		47832mc	79.9%	81.4%	
p6	80.4%	78.5%	84.9%	p6	67.7%	63.7%	75.3%	p6	79.4%	80.6%	84.7%
**Amino Acid Sequence Identity**											
**MYP**				**VXH**				**RJ**			
	rab81	rab52	47832mc		rab81	rab52	47832mc		rab81	rab52	47832mc
rab52	88.5%			rab52	81%			rab52	91.9%		
47832mc	90.9%	91.1%		47832mc	77.7%	72.8%		47832mc	92.4%	95.5%	
p6	91.2%	90.2%	97.4%	p6	79.2%	72.9%	82%	p6	91.9%	95%	99%

## Data Availability

All relevant information is contained within the manuscript or the Appendix A.

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
