# Peer review of "A Modular Hepatitis E Virus Replicon System for Studies on the Role of ORF1-Encoded Polyprotein Domains"

_pathogens, 2022, doi:10.3390/pathogens11030355_

Round 1

Reviewer 1 Report

In order to identify genomic regions responsible for HEV propagation in cell culture and to evaluate the role of different segments of the ORF1-encoded polyprotein, the authors generated Gaussia luciferase expressing full-length replicons based on human and rabbits HEV strains (47832mc, rab52 and rab81) and assembled 9 chimeric luciferase reporter replicons based on the Kernow-C1/p6 replicon backbone by partitioning ORF1 into three fragments and exchanging each of the fragments of p6 separately with the corresponding fragments from the 47832mc, rab52 and rab81 strains. Luciferase activity of rabbit HEV -derived replicons was significantly lower than that of the p6 and 47832mc replicons. Serial exchanges of three ORF1 segments within the p6 backbone indicated that HEV replication in HepG2 cells is not determined by a single domain. Based on the results obtained, the authors conclude that a specific combination of viral factors is required for efficient HEV propagation in cell culture.

The authors’ study is interesting and seems to be worthy of note. Genomic regions within ORF1 that are responsible for HEV propagation have not yet been specified and such attempts deserve further analysis. However, this manuscript has serious reservations that need to be addressed as described below.

Comments:

  1. HEV replicons harboring the Gaussia luciferase gene that is inserted in place of ORF2 of the HEV genome have been successfully constructed and utilized by many research groups (Debing et al., Antimicrob Agents Chemother 58, 267-273, 2014, Debing et al., Dis Model Mech 9, 1203-1210, 2016; Qu et al., Arch Virol 162, 2989-2996, 2017; Wang et al., Antiviral Res 140, 1-12, 2017). Similar studies to identify genomic regions responsible for HEV propagation using chimeric HEV replicons have also been reported by several research groups (Feagins et al., J Med Virol 80, 1379-1386, 2008; Cordoba et al., J Gen Virol 93, 2183-2194, 2012; Chatterjee et al., J Gen Virol, 97, 1829-1840, 2016; Tian et al., J Med Virol 92, 3563-3571, 2020). Therefore, the methods used in the present study are of little novelty. The statement “the establishment of a novel modular luciferase replicon system,,,” (line 204) is misleading and should be modified.
  2. In this study, HEV replication activity was evaluated only by the luciferase reporter assay. Mysteriously, replicon activities dropped to base line over 7 days after RNA transfection, with the peak activity on day 3 (Figs. 2 and 3). The authors should quantitate the loads of particle-associated HEV RNA in culture media in order to demonstrate that the viral loads are correlated with the luciferase activity. Newly obtained data should be presented in Figs. 2 and 3.
  3. It is incomprehensible that luciferase activities have decreased over time after peaking on day 3. To confirm the stability of the inserted Gaussia luciferase gene, the presence of the complete luciferase gene in the reporter replicons (in culture media on day 7) should be verified by PCR amplifications targeting the luciferase gene, followed by sequence analysis of the amplicons.

    In order to identify genomic regions responsible for HEV propagation in cell culture and to evaluate the role of different segments of the ORF1-encoded polyprotein, the authors generated Gaussia luciferase expressing full-length replicons based on human and rabbits HEV strains (47832mc, rab52 and rab81) and assembled 9 chimeric luciferase reporter replicons based on the Kernow-C1/p6 replicon backbone by partitioning ORF1 into three fragments and exchanging each of the fragments of p6 separately with the corresponding fragments from the 47832mc, rab52 and rab81 strains. Luciferase activity of rabbit HEV -derived replicons was significantly lower than that of the p6 and 47832mc replicons. Serial exchanges of three ORF1 segments within the p6 backbone indicated that HEV replication in HepG2 cells is not determined by a single domain. Based on the results obtained, the authors conclude that a specific combination of viral factors is required for efficient HEV propagation in cell culture.

    The authors’ study is interesting and seems to be worthy of note. Genomic regions within ORF1 that are responsible for HEV propagation have not yet been specified and such attempts deserve further analysis. However, this manuscript has serious reservations that need to be addressed as described below.

    Comments:

    1. HEV replicons harboring the Gaussia luciferase gene that is inserted in place of ORF2 of the HEV genome have been successfully constructed and utilized by many research groups (Debing et al., Antimicrob Agents Chemother 58, 267-273, 2014, Debing et al., Dis Model Mech 9, 1203-1210, 2016; Qu et al., Arch Virol 162, 2989-2996, 2017; Wang et al., Antiviral Res 140, 1-12, 2017). Similar studies to identify genomic regions responsible for HEV propagation using chimeric HEV replicons have also been reported by several research groups (Feagins et al., J Med Virol 80, 1379-1386, 2008; Cordoba et al., J Gen Virol 93, 2183-2194, 2012; Chatterjee et al., J Gen Virol, 97, 1829-1840, 2016; Tian et al., J Med Virol 92, 3563-3571, 2020). Therefore, the methods used in the present study are of little novelty. The statement “the establishment of a novel modular luciferase replicon system,,,” (line 204) is misleading and should be modified.
    2. In this study, HEV replication activity was evaluated only by the luciferase reporter assay. Mysteriously, replicon activities dropped to base line over 7 days after RNA transfection, with the peak activity on day 3 (Figs. 2 and 3). The authors should quantitate the loads of particle-associated HEV RNA in culture media in order to demonstrate that the viral loads are correlated with the luciferase activity. Newly obtained data should be presented in Figs. 2 and 3.
    3. It is incomprehensible that luciferase activities have decreased over time after peaking on day 3. To confirm the stability of the inserted Gaussia luciferase gene, the presence of the complete luciferase gene in the reporter replicons (in culture media on day 7) should be verified by PCR amplifications targeting the luciferase gene, followed by sequence analysis of the amplicons.

Author Response

Reviewer: 1

In order to identify genomic regions responsible for HEV propagation in cell culture and to evaluate the role of different segments of the ORF1-encoded polyprotein, the authors generated Gaussia luciferase expressing full-length replicons based on human and rabbits HEV strains (47832mc, rab52 and rab81) and assembled 9 chimeric luciferase reporter replicons based on the Kernow-C1/p6 replicon backbone by partitioning ORF1 into three fragments and exchanging each of the fragments of p6 separately with the corresponding fragments from the 47832mc, rab52 and rab81 strains. Luciferase activity of rabbit HEV -derived replicons was significantly lower than that of the p6 and 47832mc replicons. Serial exchanges of three ORF1 segments within the p6 backbone indicated that HEV replication in HepG2 cells is not determined by a single domain. Based on the results obtained, the authors conclude that a specific combination of viral factors is required for efficient HEV propagation in cell culture.

The authors’ study is interesting and seems to be worthy of note. Genomic regions within ORF1 that are responsible for HEV propagation have not yet been specified and such attempts deserve further analysis. However, this manuscript has serious reservations that need to be addressed as described below.

>Reply: Thank you for your review. We tried to follow all suggestions.

Comments:

  1. HEV replicons harboring the Gaussia luciferase gene that is inserted in place of ORF2 of the HEV genome have been successfully constructed and utilized by many research groups (Debing et al., Antimicrob Agents Chemother 58, 267-273, 2014, Debing et al., Dis Model Mech 9, 1203-1210, 2016; Qu et al., Arch Virol 162, 2989-2996, 2017; Wang et al., Antiviral Res 140, 1-12, 2017). Similar studies to identify genomic regions responsible for HEV propagation using chimeric HEV replicons have also been reported by several research groups (Feagins et al., J Med Virol 80, 1379-1386, 2008; Cordoba et al., J Gen Virol 93, 2183-2194, 2012; Chatterjee et al., J Gen Virol, 97, 1829-1840, 2016; Tian et al., J Med Virol 92, 3563-3571, 2020). Therefore, the methods used in the present study are of little novelty. The statement “the establishment of a novel modular luciferase replicon system,,,” (line 204) is misleading and should be modified.

>Reply: The Word “novel” in this context referred specifically to the modular assembly of replicons utilized in this work. Nevertheless, to avoid confusion, the statement was amended as requested. Additionally, the introduction has been adapted to include previous publications involving chimeric HEV constructs mentioned by the reviewer (including Cordoba et al., 2012 and Tian et al, 2020). The background on Gaussia reporter replicons had already been mentioned with p6/Luc (Shukla et al., 2012, J Virol, 2012. 86(10): p. 5697-707) which, as we have stated, is the gold standard which our work is based on. As noted by the reviewer, the same system has also been used by other groups, including (Debing et al., Antimicrob Agents Chemother 58, 267-273, 2014, Debing et al., Dis Model Mech 9, 1203-1210, 2016; Qu et al., Arch Virol 162, 2989-2996, 2017; Wang et al., Antiviral Res 140, 1-12, 2017).

  1. In this study, HEV replication activity was evaluated only by the luciferase reporter assay. Mysteriously, replicon activities dropped to base line over 7 days after RNA transfection, with the peak activity on day 3 (Figs. 2 and 3). The authors should quantitate the loads of particle-associated HEV RNA in culture media in order to demonstrate that the viral loads are correlated with the luciferase activity. Newly obtained data should be presented in Figs. 2 and 3.

>Reply: There appears to have been a misunderstanding of the methods we utilized. The replicons employed in this work are based on the architecture of p6/luc. They cannot express a functional capsid gene and therefore cannot form infectious particles (Shukla et al., 2012, J Virol, 2012. 86(10): p. 5697-707). The reason for that is that the reporter gene is inserted at the start codon of the capsid protein coding sequence (CDS) and contains a stop codon, which prevents translation of the remaining sequence downstream of the luciferase CDS. Additionally, this modification deletes the coding sequence of the N-terminal domain of the capsid protein, which is required for interaction of the capsid protein with the viral RNA. Therefore, the possibility of infectious virion formation can be excluded in any case. This is one of the major advantages of the luciferase assay, which reflects the translation activity of the subgenomic RNA and therefore the activity of the ORF1 protein without the need for effective infection, which is always a challenge with cell culture derived HEV. The method is well established and is commonly used in the field. Furthermore, the activity peak followed by a decline of luciferase expression has been demonstrated even in the initial description of p6/luc replicon (Shukla et al., 2012, J Virol, 2012. 86(10): p. 5697-707) and is attributed directly to the lack of a capsid protein. In this case, the peak occurred around day 6, a bit later than in our work. This difference is likely due to the choice of host cell line (HepG2 vs S10-3) and transfection protocol (electroporation vs lipid-based transfection). Other authors have performed similar experiments by electroporating HepG2 cells, as we did, and showed high and/or peaking expression levels by day 3 (e.g. Nguyen et al, 2014, Journal of Virology 88 (2): 868–77.; Knegendorf et al., 2018, Hepatology Communications 2 (2): 173–87.; Ding et al., 2018, MBio 9 (3)). With the exception of Nguyen et al, the assays were not continued beyond that point, presumably due to reaching the expression peak.

Corresponding explanations were added ad the last paragraph of the introduction (lanes 79-112), the first paragraph of Results section 2.1 (lanes 145-153), Figure 1 was adapted and a short passage was added to the discussion (lanes 282-291) in order to emphasize this property of the reporter replicons.

  1. It is incomprehensible that luciferase activities have decreased over time after peaking on day 3. To confirm the stability of the inserted Gaussia luciferase gene, the presence of the complete luciferase gene in the reporter replicons (in culture media on day 7) should be verified by PCR amplifications targeting the luciferase gene, followed by sequence analysis of the amplicons.

>Reply: As mentioned above, the peak and subsequent decrease of luciferase activity is expected behavior of non-infectious luciferase reporter replicons (see Reply to comment 2). Sequencing the RNA in the culture media would not be productive, since no virus particles are secreted into the supernatant. We do agree that in-depth investigation on the factors involved in this long-term decline of replicon activity would be very interesting, particularly because this could potentially shed some light on mechanisms of viral clearance. However, that is well beyond the scope of this manuscript. 

Reviewer 2 Report

The manuscript by Cierniak F et al entitled “A modular hepatitis E virus replicon system for studies on the role of ORF1-encoded polyprotein domains”, presents an original study in which the authors used sequence diversity between distantly related HEV3 to study the viral RNA replication machinery. Based on subgenomic replicon constructs allowing the expression of Gaussia luciferase as a reporter, they swapped large portions of the ORF1 within a highly replicating well-established molecular clone, namely p6. Interestingly, swapping of domains from rabbit HEV isolates of which the entire replicon was not replicating or at very low level, led to significant replication in the HEV p6 background. This study provides new information on the important domains of ORF1 for the replication. The study is well-controlled and used state-of-the-art techniques. 

Minor issues:

- The overall percentage of amino acid and nucleotide identities between the different strains may be indicated as well in Table 1. Moreover, a sequence alignment of the different domains may be provided in Supplementary material.

- Figure 3 may be oriented in portrait view with panel A reduced in size to give more focus to the results presented in panel B. In addition, to improve the understanding of the figure, I would suggest to indicate in the domain swapped (coloured), the percentage of identity with p6. It may allow to directly correlate the luciferase results shown in B with the variation in sequence. Therefore, the authors may simply remove the MT, Y, PCP, … labelling into each portion but keep it for the wt construct shown on the top.  

- The authors may want to revise the Table 1 as the fomatting of the table may have been modified which renders it difficult to read.

- The discussion section appears long and may be streamlined. In particular, the second paragraph which does not bring crucial element of discussion, may be removed. 

Author Response

Reviewer: 2

The manuscript by Cierniak F et al entitled “A modular hepatitis E virus replicon system for studies on the role of ORF1-encoded polyprotein domains”, presents an original study in which the authors used sequence diversity between distantly related HEV3 to study the viral RNA replication machinery. Based on subgenomic replicon constructs allowing the expression of Gaussia luciferase as a reporter, they swapped large portions of the ORF1 within a highly replicating well-established molecular clone, namely p6. Interestingly, swapping of domains from rabbit HEV isolates of which the entire replicon was not replicating or at very low level, led to significant replication in the HEV p6 background. This study provides new information on the important domains of ORF1 for the replication. The study is well-controlled and used state-of-the-art techniques.

>Reply: Thank you for your review. The manuscript has been revised accordingly. 

Minor issues:

- The overall percentage of amino acid and nucleotide identities between the different strains may be indicated as well in Table 1. Moreover, a sequence alignment of the different domains may be provided in Supplementary material.

>Reply: As suggested, a table depicting the overall percent identities of the ORF1-encoded proteins has been added (Table 1). Amino acid sequence alignments of the exchanged ORF1-encoded segments are now included in the supplementary material (Figures S1-S3).

- Figure 3 may be oriented in portrait view with panel A reduced in size to give more focus to the results presented in panel B. In addition, to improve the understanding of the figure, I would suggest to indicate in the domain swapped (coloured), the percentage of identity with p6. It may allow to directly correlate the luciferase results shown in B with the variation in sequence. Therefore, the authors may simply remove the MT, Y, PCP, … labelling into each portion but keep it for the wt construct shown on the top. 

>Reply: Figure 3 has been adapted as requested.

- The authors may want to revise the Table 1 as the fomatting of the table may have been modified which renders it difficult to read.

>Reply: We will have to consult the editor on this problem. Thank you for the notification.

- The discussion section appears long and may be streamlined. In particular, the second paragraph which does not bring crucial element of discussion, may be removed.

>Reply: As suggested, the discussion was shortened overall, by deleting the second (lanes 305-317), and second to last paragraphs (lanes 396-412). A short passage was added in response to reviewer 1.

Round 2

Reviewer 1 Report

The manuscript has been revised properly.